# Sex-based differences in long-term outcomes after stroke: A meta-analysis

**Xiumei Guo[1,2]☉, Yu Xiong[2]☉, Xinyue Huang[2]☉, Zhigang Pan[2], Xiaodong Kang[2], Chunhui Chen[2], Jianfeng Zhou[2], Hanlin Zheng[2], Yuping Chen[2]\*, Weipeng Hu[2]\*, Lingxing Wang[1]\*, Feng Zheng[2]\***

1 Department of Neurology, the Second Affiliated Hospital, Fujian Medical University, Quanzhou, Fujian Province, China, 2 Department of Neurosurgery, the Second Affiliated Hospital, Fujian Medical University, Quanzhou, Fujian Province, China

☉ These authors contributed equally to this work.
\* dr.feng.zheng@gmail.com (FZ); lxing502@fjmu.edu.cn (LW); 1923307274@qq.com (WH); 503106356@qq.com (YC)

**Data Availability Statement:** All relevant data are within the paper and its Supporting Information files.

**Funding:** The author(s) received no specific funding for this work.

## Abstract

### Background

There is limited data on sex-related disparities in the long-term outcomes after stroke. We aim to investigate whether there are sex-based differences in long-term outcomes using pooled data.

### Methods

Three databases (PubMed, Embase, and Cochrane Library) were systematically searched from inception to July 2022. This meta-analysis was performed in accordance with the recommendations and guidelines of the Preferred Reporting Items for Systematic Reviews and Meta-Analyses. The modified Newcastle-Ottawa scale was used to assess the risk of bias. In addition, a random-effects model was used.

### Results

Twenty-two cohort studies with 84538 patients were included. There were 50.2% men and 49.8% women. Women had a higher mortality at 1 (odds ration [OR], 0.82; 95% confidence interval [CI][0.69, 0.99], P = 0.03) and 10 (OR 0.72, 95% CI[0.65, 0.79], P < 0.00001) years, higher stroke recurrence at 1 year (OR 0.85, 95% CI[0.73, 0.98], P = 0.02), lower favorable outcome at 1 year (OR 1.36, 95% CI[1.24, 1.49], P < 0.00001). No significant difference was detected between men and women in the outcomes of health-related quality of life and depression.

### Conclusion

In this meta-analysis, the 1- and 10-year mortality and stroke recurrence rates were higher in female patients than in male patients after stroke. In addition, females tended to experience less favorable outcomes in the first year after stroke. Finally, further long-term studies

**Competing interests:** The authors have declared that no competing interests exist.

on sex disparities in stroke prevention, care, and management are warranted to explore the opportunities to reduce this gap.

## 1. Introduction

Stroke affects female and male patients differently, although the reasons and mechanisms of these differences are unclear [1, 2]. It is known that there are sex-based differences in various factors of stroke, including risk factors, clinical acute stroke symptoms, and treatment management [3]. Moreover, previous studies have reported sex-based differences in stroke outcomes, where females are more likely to have greater stroke severity and poorer outcomes than males [4–7]. These findings are largely based on studies that have assessed stroke outcomes at discharge or within the first few months post-onset [4, 5, 7]. However, data on the long-term outcomes after stroke remain scarce [8, 9].

Although reductions in stroke mortality have occurred in recent years [10], females continue to bear a disproportionate burden of stroke compared to males, and this remains a significant global concern [11]. Understanding sex-based differences in stroke epidemiology and outcomes is important for reducing potential disparities and the excess burden of disability [8]. However, despite increasing evidence and interest in sex-based disparities in health, the available data on sex-related differences in long-term outcomes after stroke are limited [8, 9]. Therefore, this meta-analysis was performed to compare various long-term stroke outcomes between males and females to investigate whether there is a sex-based disparity in long-term stroke outcomes and provide insights for improving stroke care and management.

## 2. Methods

### 2.1. Search strategy

This meta-analysis was performed in accordance with the recommendations and guidelines of the Preferred Reporting Items for Systematic Reviews and Meta-Analyses (PRISMA) (S1 Checklist) [12]. Two authors (X.G. and F.Z.) independently performed a comprehensive literature search. Studies were retrieved from the PubMed, Embase, and Cochrane databases from inception to August 2022, with no language restrictions. The complete search strategy is provided in the Supplementary Materials section (S1 Text in S1 File). Additionally, the reference lists of the included studies were manually searched to identify additional relevant publications. All studies that met the inclusion criteria, as described below, were included in the analysis. Any discrepancies discovered during the literature search were resolved through consultation with the corresponding author (F.Z.).

### 2.2. Outcomes

Death, stroke recurrence, and favorable outcomes were the primary outcomes. Death was defined as all-cause mortality. Stroke recurrence was self-reported and defined as the aggravation of the original symptoms, the appearance of new signs, or re-hospitalization with a hospital admission diagnosis of recurrent stroke. Favorable outcomes referred to an individual's ability to perform activities of daily living. In stroke, this outcome is closely related to the severity and type of neurological impairments. The modified Rankin scale (mRS) was used to assess functional outcomes and independent living [13]. The mRS is an ordinal scale ranging from 0–6, with 0 indicating no symptoms and 6 indicating patient death. A favorable outcome

was defined as an mRS score of 0–2. Restoring favorable outcomes is a focus of stroke rehabilitation as they are strongly associated with independent living.

The secondary outcomes included health-related quality of life (HRQoL) and depression. The World Health Organization defines "quality of life" as an "individual's perception of their position in life in the context of the culture and value systems in which they live in relation to their goals and expectations" [14]. HRQoL instruments commonly comprise the physical, social, and mental domains, aligning with the notion that health is a state of complete well-being and not just an absence of disease. The 36-Item Short Form Health Survey (SF-36) is the most commonly used instrument to measure HRQoL [15]. Depression was defined as a Hospital Anxiety subscale score of ≥11 [16].

## 2.3. Study selection

Two researchers (X. G. and Y. X.) independently screened the retrieved literature. The inclusion criteria for selecting the studies were as follows: (a) randomized controlled trials or observational cohort studies; (b) studies comparing long-term stroke outcomes among female and male patients with stroke; and (c) studies that reported at least one outcome. Exclusion criteria were as follows: (a) studies that used non-human subjects; (b) studies from which data could not be extracted; (c) non-comparative studies, including case reports, reviews, conference abstracts, letters, surveys, or satisfaction studies; (d) studies only equipped with a single-arm design; and (e) studies that did not meet the inclusion criteria.

## 2.4. Data extraction

Two authors (X.G. and Y.X.) independently extracted the baseline characteristics and primary and secondary outcomes from the included studies. Baseline data included the first author, year of publication, sample size, study design (hospital-based or population-based), age, the subtype of stroke, potential risk factors, stroke severity, and pre-stroke health. Any questions were communicated and resolved by the corresponding author (F.Z.).

## 2.5. Critical appraisal

Two authors (X. G. and Y. X.) independently assessed the quality of the included studies. Cochrane risk-of-bias tool for randomized trials (RoB 2) was used to assess the bias in randomized controlled studies [17]. The degree of bias risk (low, unknown, and high) was evaluated across seven aspects (random sequence generation, allocation concealment, blinding of researchers and participants, the integrity of outcome data, blind evaluation of research outcomes, selective reporting of research results, and other sources of bias) to reflect the quality of each study. The Newcastle Ottawa Scale, which assigns a score out of a possible of nine stars, was used to assess the quality of the observational cohort studies included in this meta-analysis [18]. The Scale rates were studied under the selection, comparability, and outcome categories. Four stars were available in the selection category, two in the comparability category, and three possible stars could be achieved in the outcome category. The maximum number of stars was nine, and studies were graded as "high quality," with scores of ≥6, "moderate" with scores of 4–5, and "weak," with scores of 0–3. A Newcastle Ottawa Scale score >7 indicated high quality. The quality of the included studies was evaluated carefully, and any differences in opinions were resolved through discussion.

## 2.6. Statistical analysis

This study assessed the differences between males and females in the following outcomes: mortality, favorable outcomes, stroke recurrence, HRQoL, and depression. Categorical variables were analyzed using the dominance odds ratio (OR), and continuous variables were analyzed using the mean difference (MD). In addition, a random effects model was used to pool the results of the included studies.

Statistical analysis was performed using the Review Manager software (version 5.4; The Cochrane Collaboration,2020; Nordic Cochrane Center, Copenhagen, Denmark). Dichotomous data were summarized using odds ratios (OR) and 95% confidence intervals (CI). Continuous data are displayed as MD and standard deviation (SD). Where appropriate, the SD was calculated based on the reported standard errors. Statistical significance was set at *P*-values < 0.05. Statistical heterogeneity was measured using $c^2$ and $I^2$ statistics. According to the recommendation of the Cochrane Statistical Methods Group [19], a significance level of heterogeneity was set at a *P*-value of 0.1, and the $I^2$ statistic was interpreted as follows: 0–40%, low heterogeneity; 30–60%, moderate heterogeneity; 50–90%, substantial heterogeneity; and 75–100%, considerable heterogeneity. Statistically significant heterogeneity was present at $P <$ .1 and $I^2 > 50\%$. In these cases, sensitivity analysis and subgroup analysis were performed to assess the robustness of the results.

# 3. Results

## 3.1. Study inclusion

A total of 1636 publications were retrieved, of which 1452 were obtained after removing duplicate studies. After screening titles and articles, a total of 1225 articles were excluded. The full text of the remaining 227 articles was assessed for eligibility, resulting in the exclusion of 150 studies with outcomes not stratified by sex and 37 conference abstracts without a complete text. Three studies were excluded because their data were not extractable. Fifteen of the remaining articles were excluded due to a lack of long-term outcome data based on sex-based differences. In total, 22 studies [8, 9, 20–39] comprising a total of 84538 patients (42431 and 42107 in the male and female groups, respectively) comparing long-term clinical outcomes after stroke were included in the present analysis. A flowchart of the search strategy is shown in Fig 1.

Until the literature search, no randomized controlled trials had been published on this topic. Hence, all 22 [8, 9, 20–39] articles were cohort studies (14 hospital-based studies and 8 population-based studies). The details of these studies are summarized in Table 1.

## 3.2. Primary outcomes

**3.2.1. Mortality.** First-year mortality was recorded in 13 studies [9, 26–31, 35–37, 39–41]. Pooled results showed a significant difference in first-year mortality between male and female stroke patients, in favor of male patients (odds ratio [OR], 0.82; 95% confidence interval [CI] 0.69 to 0.99%; *P* = .03) (Fig 2A). Since the data based on ischemic stroke were sufficiently provided in eight studies [9, 26, 29–31, 36, 39, 41], further subgroup analysis was performed on the outcome of the 1-year mortality, with no significant difference detected in the ischemic stroke subgroup (OR 0.83, 95% CI 0.67% to 1.03%, *P* = .09) (S1 Fig in S1 File). Due to the substantial heterogeneity in the assessments of above two outcomes ($I^2$ = 90%, P<0.00001 and $I^2$ = 92%, P<0.00001), a sensitivity analaysis was performed, without having detected the exact heterogeneity source. Four studies [8, 20, 21, 27] recorded data assessing mortality between males and females over 5 years. There was no significant difference between the two groups

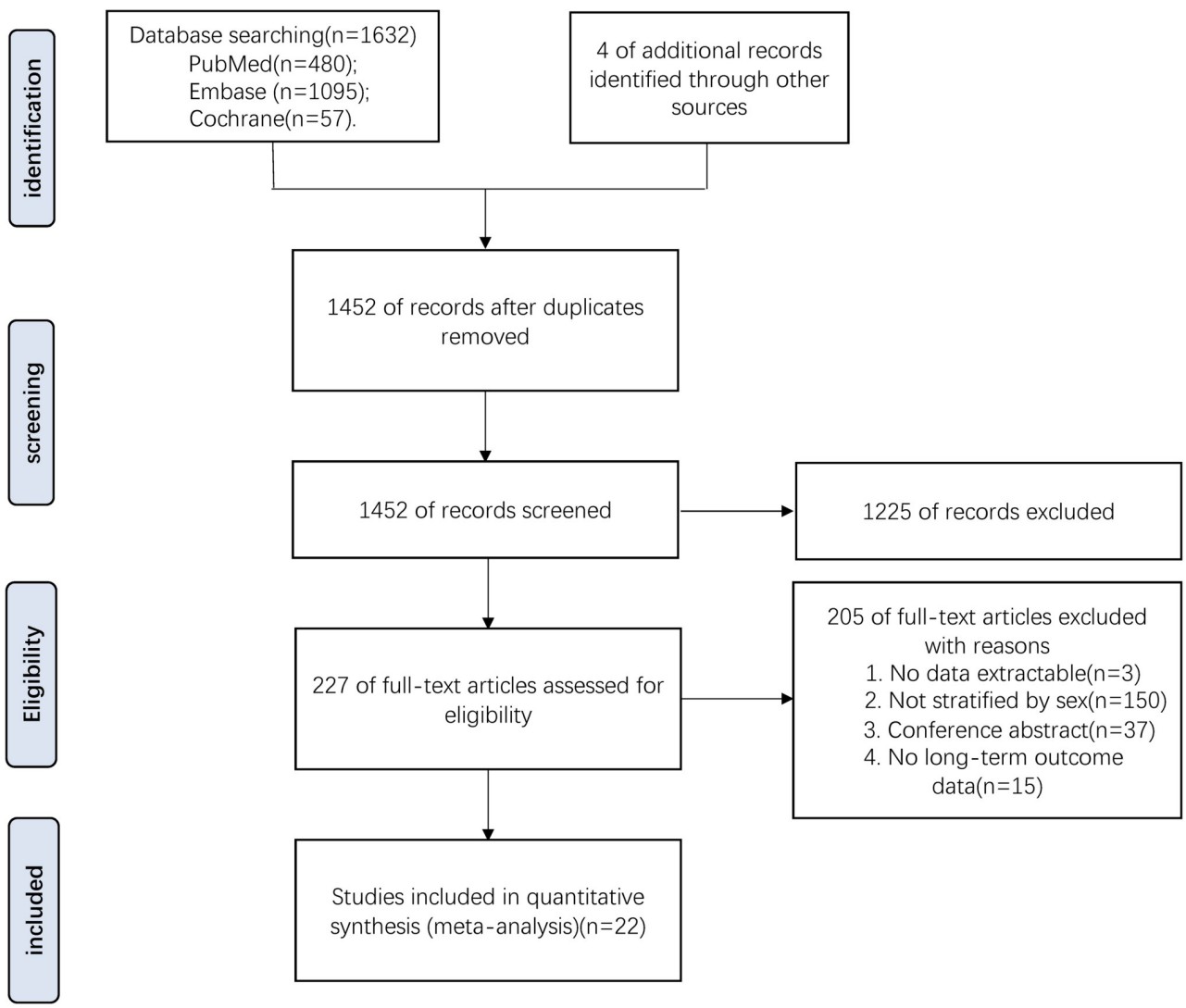

**Fig 1. Flow chart showing the search strategy.**

(OR 0.87, 95% CI 0.57% to 1.32%, $P$ = .52) (Fig 2B). Due to substantial heterogeneity ($P$ = .003, $I^2$ = 78%), a sensitivity analysis was employed to assess the robustness of the findings. After excluding one study [20], significant differences were detected in the pooled results (OR 0.68, 95% CI 0.63% to 0.75%, $P$ < .00001), without substantial heterogeneity ($P$ = .98, $I^2$ = 0%). The 10-year mortality was recorded in two studies [8, 34], with a significant difference in favor of males (OR 0.72, 95% CI 0.65% to 0.79%, $P$ < .00001) (Fig 2C).

**3.2.2. Favorable outcome.** Four of the included studies [9, 27–29] investigated favorable outcomes 1 year after stroke. Compared with males, females had significantly lower favorable outcomes (OR 1.36, 95% CI 1.24% to 1.49%, $P$ < .00001) (Fig 2D). Further subgroup analysis of patients with ischemic stroke showed a greater tendency for 1-year favorable outcomes in male patients than in female patients (OR 1.36, 95% CI 1.23% to 1.50%, $P$ < .0001) (S1 Fig in S1 File). Regarding 5-year favorable outcomes, no significant difference was detected between males and females (OR 2.88, 95% CI 0.21% to 38.63%, $P$ = .43) (Fig 2E).

**Table 1. Baseline characteristic for included studies.**

| Author and publication year | Study design and period | Total (n) | Patients(n) M F | Age M F | Subtype(%) M F | Stroke severity† M F | Smoke (%) M F | Hypertension (%) M F | Diabetes (%) M F | AF (%) M F | IHD (%) M F | Dyslipidemia (%) M F | Drink (%) M F | Pre-stroke status§ M F | NOS |
|---|---|---|---|---|---|---|---|---|---|---|---|---|---|---|---|
| Akhtar 2020 (21) | H-B 2014–2019 | 819 | 519 372 | 62.9±14.1 65.9 ±14.2 | IS 71.7 IS 70.2 ICH 13.7 ICH 13.7 TIA 14.6 TIA 16.1 | 2(0–4) 1(0–3) | 32.4 0.8 | 83.4 82.5 | 68.2 74.7 | 8.9 14.0 | 20.2 11.8 | 51.8 55.1 | NA NA | NA NA | 7 |
| Feigin 2010 (24) | P-B 2007–2008 | 418 | 217 201 | 64.1±4.1 75.5±2.5 | IS 80.4* ICH 8.1 UND 5.0 | 4.1(4.3) 4.7(4.9) | NA NA | NA NA | NA NA | NA NA | NA NA | NA NA | NA NA | NA NA | 8 |
| Geng 2019 (26) | H-B 2004–2008 | 228 | 147 81 | 42.78±6.0 42.3 ±6.6 | IS 100 IS 100 | 4(2,10) 4(1,8) | 51.7 5.0 | 34.0 19.8 | 8.8 5.0 | 5.4 14.8 | 0.7 1.0 | 5.4 1.0 | 37.4 2.5 | NA NA | 7 |
| Kim 2010 (28) | H-B 2005–2009 | 1055 | 575 480 | 64.83±11.98 70.09 ±13.02 | IS 87.2 IS 86.9 TIA 5.7 TIA .8 ICH 7.1 ICH8.3 | Mild 87.1 Mild 82 Moderate10.2 Moderate7.7 Severe7.3. Severe5.2 | 46.9 6.8 | 60.1 62.6 | 33.3 29.2 | NA NA | NA NA | 16.1 22.5 | NA NA | NA NA | 8 |
| Prencipe 1998 (34) | H-B 1077–1986 | 322 | 244 78 | 55* | NA NA | NA NA | NA NA | 54* | 17* | NA NA | NA NA | 42 NA | NA NA | NA NA | 8 |
| Ridder 2016 (23) | H-B 2006–2014 | 823 | 443 380 | 70(12) 73(12) | ICH 100 ICH 100 | 12(5–26) 15(6–28) | NA NA | 83 78 | NA NA | 24 19 | NA NA | NA NA | NA NA | 0(0–2) 1(0–2) | 8 |
| Sarfo 2017 (35) | H-B 2012–2014 | 607 | 305 302 | 59.9±13,9* | IS 40.3* ICH 18.6 UND 41.1 | NA NA | NA NA | NA NA | NA NA | NA NA | NA NA | NA NA | NA NA | NA NA | 7 |
| Xu 2022 (8) | P-B 1995–2019 | 6687 | 3438 3249 | 0-64y:42 0-64y:26 65-74y:26 65-74y:21 75-84y:27 74-84y:23 >85y:9 >85y:710 | IS 70 IS 71 ICH 13 ICH 11 SH 4 SH 5 UNC 2 UNC 4 UND 10 UND 9 | Mild 28 Mild 22 Moderate 29 Moderate 31 Severe 6 Severe 8 Unknown 37 Unknown 40 | 66 42 | 64 67 | 22 21 | 15 20 | 20 19 | 30 28 | NA NA | 5 12 | 9 |
| Wang 2022 (9) | P-B 2015–2018 | 9038 | 6236 2802 | 61.26±11.42 64.78 ±10.84 | IS 100 IS 100 | 4.43±4.2 4.85±4.44 | 45.1 3.43 | 68.13 60.89 | 22.77 27.09 | 6.59 9.74 | 9.77 13.13 | 8.07 8.57 | NA NA | 0.42±0.88 0.44±0.92 | 9 |
| Adoukonou 2020 (20) | H-B 2012–2018 | 247 | 136 111 | 58.1±13.4* | IS 36.8* ICH 27.9 UND 35.2 | NA NA | NA NA | NA NA | NA NA | NA NA | NA NA | NA NA | NA NA | NA NA | 6 |
| Johnson 2015 (27) | H-B 2008–2009 | 155 | 94 61 | 79.5±8.7 84.8±8.1 | ICH 85.2* | Mild 59 Mild 47 Moderate16 Moderate17 Major 25 Major 36 | 63 25 | 61 65 | 31 15 | NA NA | NA NA | NA NA | NA NA | NA NA | 7 |
| Vemmons 2000 (38) | P-B 1991–1992 | 152 | 44 108 | NA NA | IS 64.9 IS 75,7 ICH 20.9 ICH 12.5 PAD 14.2 PAD11.8 | 2(0–6) 2(0–5) | 34 2.1 | 78.4 84.6 | 27.9 32.2 | 30.8 38.3 | 10.4 7.0 | 41.1 46.4 | 20.8 1.4 | NA NA | 7 |
| Kong 2010 (29) | H-B 2002–2007 | 2774 | 1655 1119 | 63±13.6 65±13.5 | IS 100 IS 100 | 7.32 8.72 | 44.4 5.0 | 56.3 56.6 | 17.7 21.7 | 10.9 29.7 | 1.6 2.3 | 11.5 9.6 | 30.6 3.0 | NA NA | 8 |
| Cordato 2005 (22) | H-B 2002–2003 | 186 | 90 96 | 65–74.9y:15.4* 75–84.9y:28.4 ≥85y:48.9 | NA NA | NA NA | NA NA | 30.5* | 30.3* | 42.4* | 29.2* | NA NA | 14.4 6 | 25.2 NA | 7 |
| Mohan 2009 (32) | P-B 1995–2004 | 2874 | 1427 1447 | < 65y:30.0* 65-74y:26.6 75-84y:43.4 | IS 72.7* ICH 13.7 SH 6.0 UND 7.6 | GCS<8: 18.1* GCS 9–12: 15.3 GCS 13–15: 66.6 | 35.6* | 64.1* | 18.0* | 17.2* | 11.7* | NA NA | NA NA | NA NA | 7 |
| Medlin 2020 (31) | P-B 2003–2016 | 3993 | 2236 1757 | 69.9(20.4) 77 (18.1) | IS 100 IS 100 | 6(10) 7(13) | 28 18 | 71 72.4 | 21.3 15.3 | 25.5 33.7 | NA NA | NA NA | NA NA | 0(1) 1(2) | 8 |
| Carod-Artal 2000 (22) | H-B 1996–1997 | 118 | 65 53 | 65.16(11.4) 71.80(8.88) | IS 89* ICH 11 | NA NA | 28.89* | 65.56* | 30* | 26.67* | | 40 NA | NA NA | NA NA | 6 |
| Gall 2010 (25) | P-B 1995–1999 | 494 | 249 245 | 70.6(12.6) 73.7 (14.4) | IS 85.9 IS 84.9 ICH 12.1 ICH 8.2 SH 0.0 SH 0.4 UND 2.0 UND 6.5 | 4.8(5.1) 5.6(5.7) | 67 35 | 51.0 59.6 | 18.9 15.9 | 15.7 17.1 | 16.9 9.0 | NA NA | 79.5 51.6 | 19.6(1.5) 19.1(2.7) | 8 |
| Patel 2007 (33) | P-B 1995–1007 | 397 | 212 185 | < 65y:34* 65-75y:33.5 >75y:32.5 | NA NA | NA NA | NA NA | 71.8* | 62* | 18.4* | 11.6* | NA NA | NA NA | BI<15 1.5* BI 15–20 98.5 | 8 |
| Li 2015 (30) | H-B 2009–2011 | 881 | 488 383 | NA NA | IS 100 IS 100 | Mild 58.7 Mild 56.1 Moderate 27 Moderate 29.8 Severe 14.3 Severe 14.1 | 28.3 12.8 | 69.1 76.5 | 26.2 33.2 | 15.4 18.3 | NA NA | 18.9 27.2 | 11.3 0.3 | NA NA | 7 |
| Sheikh 2007 (36) | P-B 1994–1997 | 40450 | 16391 24059 | 73.5 75.5 | IS 57.1 IS 55.1 ICH 9.16 ICH 9.12 UD 32.4 UD 33.7 | NA NA | NA NA | NA NA | NA NA | NA NA | NA NA | NA NA | NA NA | NA NA | 9 |

(*Continued*)

**Table 1.** (Continued)

| Author and publication year | Study design and period | Total (n) | Patients(n) M | F | Age M | F | Subtype(%) M | F | Stroke severity† M | F | Smoke (%) M | F | Hypertension (%) M | F | Diabetes (%) M | F | AF (%) M | F | IHD (%) M | F | Dyslipidemia (%) M | F | Drink (%) M | F | Pre-stroke status§ M | F | NOS |
|---|---|---|---|---|---|---|---|---|---|---|---|---|---|---|---|---|---|---|---|---|---|---|---|---|---|---|---|
| Wang 2013 (39) | H-B 2007–2008 | 11560 | 7118 | 4442 | 64±12.3 | 67.9 ±11.9 | NA | NA | Mild 52.3 Mild 45.3 Moderate 27 Moderate 29.8 Severe 10.5 Severe 5.5 | | 60.7 | 6.0 | 61.5 | 68.1 | 19.7 | 24.8 | 5.3 | 11.1 | NA | NA | 11.5 | 11.0 | 14.9 | 0.5 | 34.5 | 34.0 | 7 |

M, male; F, female; H-B, hospital-based; P-B, population-based; AF, atrial fibrillation; IHD, ischemic heart disease; IS, ischemic stroke; ICH, intracerebral hemorrhage; SH, subarachnoid hemorrhage; UND, undetermined; UNC, unclassified; GCS, Glasgow Coma Scale; NIHSS, National Institutes of Health Stroke Scale; BI, Barthel Index.

NOS: Newcastle Ottawa scale (NOS), which gives a score out of a possible total of 9 stars. Scale rates were studied under the categories of selection, comparability, and outcome. Four stars are available in the selection category, two stars are available in the comparability section, and three possible 3 stars can be achieved in the outcome categories.

* No separate value.

† Stroke severity was measured using GCS (Mohan) or NIHSS (remaining studies).

§ Pre-stroke status was measured using BI (Patel, Gall) or mRS (remaining studies)

**3.2.3. Recurrent stroke.** Six studies [9, 25, 27, 30, 32, 39] reported the outcome of 1-year recurrent stroke, with significant differences detected in 1-year stroke recurrence between males and females (OR 0.85, 95% CI 0.73% to 0.98%, $P$ = .02) (Fig 2F) in favor of males. Subgroup analysis of ischemic stroke showed no significant difference in 1-year recurrent stroke between males and females (OR 0.98, 95% CI 0.75% to 1.28%, $P$ = .87) (S1 Fig in S1 File). The recurrence rate did not differ between males and females in the 5-year recurrence rates (OR 1.09, 95% CI 0.83% to 1.43%; $P$ = .53) (Fig 2G) and in the 10-year stroke recurrence rates (OR 0.96, 95% CI 0.76% to 1.21%, $P$ = .72) (Fig 2H).

## 3.3. Second outcomes

**3.3.1. HRQoL.** Two studies [24, 33] reported HRQoL outcomes, with no significant difference detected between male and female patients in physical health (OR -2.18, 95% CI -12.18% to 7.81%, $P$ = .67) (Fig 3A) and mental health (OR 0.39, 95% CI -2.35% to 3.13%, $P$ = .78) (Fig 3B).

**3.3.2. Depression.** The incidence of depression was reported in three studies [8, 22, 25]. No significant difference was detected between male and female patients with stroke (OR 0.39, 95% CI -2.35% to 3.13%, $P$ = .78) (Fig 3C).

## 4. Discussion

Our meta-analysis revealed several clinically relevant findings. First, regarding long-term outcomes after stroke, female patients had significantly higher 1-year and 10-year mortality rates and higher recurrence rates in the first year after stroke. In addition, females were less likely to achieve favorable outcomes in their first year. There was no significant difference in self-reported outcomes, including HRQoL and depression, as indicated by the analysis (Fig 4).

There was no significant difference in 5-year mortality between male and female patients, but there was substantial heterogeneity. Sensitivity analysis was employed according to the Cochrane Handbook for Systematic Reviews of Interventions [17] to obtain robust results. After removing the study by Adoukonou et al. [20], the heterogeneity significantly decreased, with higher 5-year mortality detected in female patients. This may be due to the high proportion of patients (over 17%) who were lost to follow-up in the study by Adoukonou et al., which may have biased the findings.

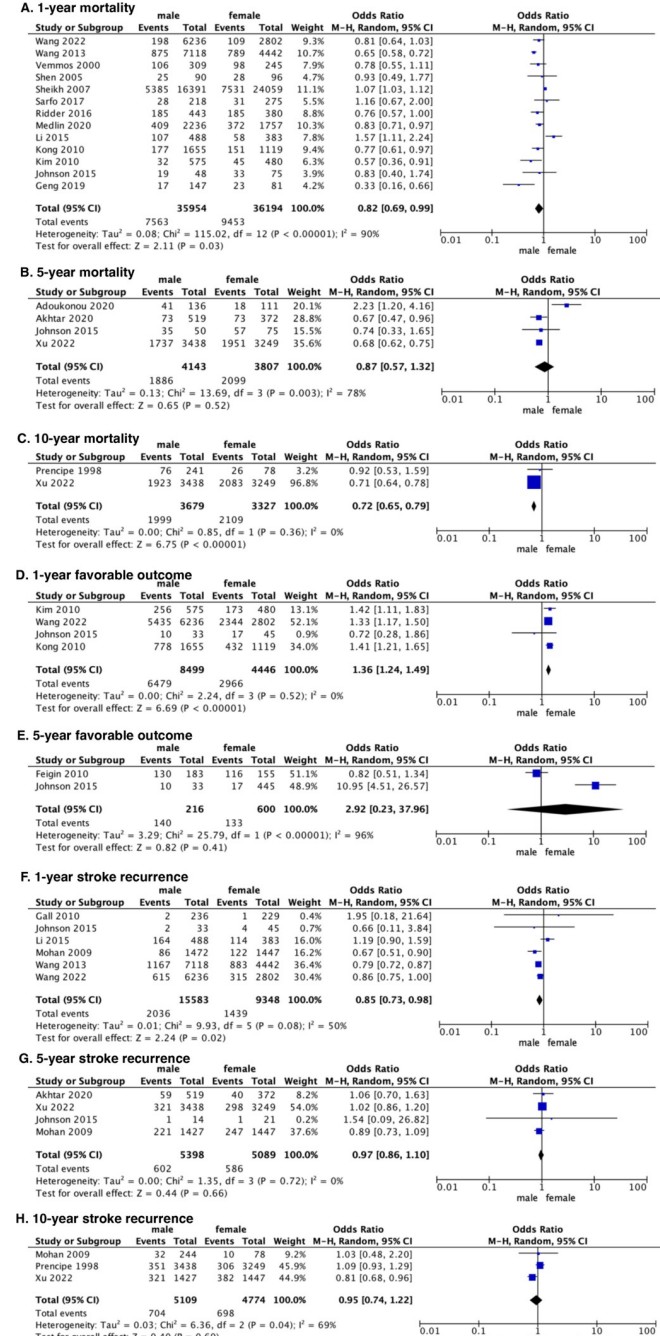

**Fig 2. Forest plots showing primary outcomes between males and females.** M-H Mantel-Haenszel statistic.

Several factors are associated with mortality. In this meta-analysis, female patients with stroke tended to be older, had higher pre-stroke handicaps, and were more serious, with higher National Institutes of Health Stroke Scale (NIHSS) scores. The studies by Xu et al. and Wang et al. [8, 9] showed that although female patients were associated with statistically higher mortality, after adjusting for relevant confounders, including age, stroke severity, stroke subtype, and other comorbidities, mortality in male patients may be higher. Among these factors,

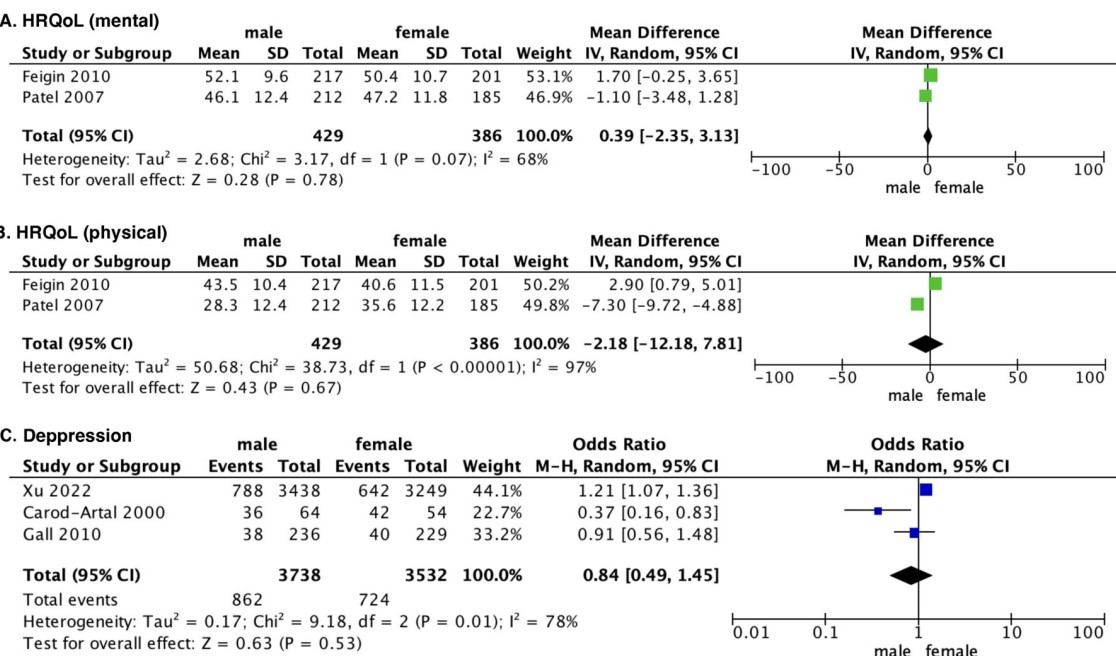

**Fig 3. Forest plot showing secondary outcomes between males and females.** M-H: Mantel-Haenszel statistic.

Wang et al. [9] believe that age may be one of the most important causes since a significant difference showed that females had lower mortality rates than males of older age. In our meta-anlaysis, the conclusion that female patients with stroke had significantly higher 1-year mortality rates should be interpreted with caution because of the significant heterogeneity. However, owing to the lack of sufficient data based on different ages, further subgroup analysis could not be performed in the present study. In addition, the subgroup analysis of ischemic stroke showed comparable mortality between males and females in the first year after stroke.

Compared with female patients, more favorable outcomes were observed in male patients 1 year after stroke, which became insignificant 5 years after stroke. A subgroup analysis was performed considering the significant influence of the stroke subtype. Pooled data regarding ischemic stroke showed that male patients had more favorable outcomes at 1 year, which was consistent with the primary outcome in the present study. Preclinical and clinical evidence suggests that a combination of factors, including exposure to sex hormones, sex chromosomes, and brain and vascular microenvironments, may contribute to sex-based differences in the cellular mechanisms underlying stroke injury [42]. Furthermore, a study by Wang et al. [9] showed that older females, especially those >65 years of age, were more prone to have poor outcomes. This may be explained by age-dependent differences between females and males, such as atrial fibrillation. The risk of atrial fibrillation-associated stroke increased with age and was higher in females than in males [43].

The present study detected a significant difference in 1-year stroke recurrence in favor of male patients. However, this difference became insignificant after 5 and 10 years. Moreover, it was reported that there was no significant difference in stroke recurrence when adjusting for confounders such as age, pre-stroke dependency, and stroke severity [8, 39]. Interestingly, in a recent study using a crude model, a significant difference in the outcome of one-year stroke recurrence was detected in male patients [9], which was consistent with our findings. However, this difference was attenuated when accounting for blood pressure and serum-related covariates.

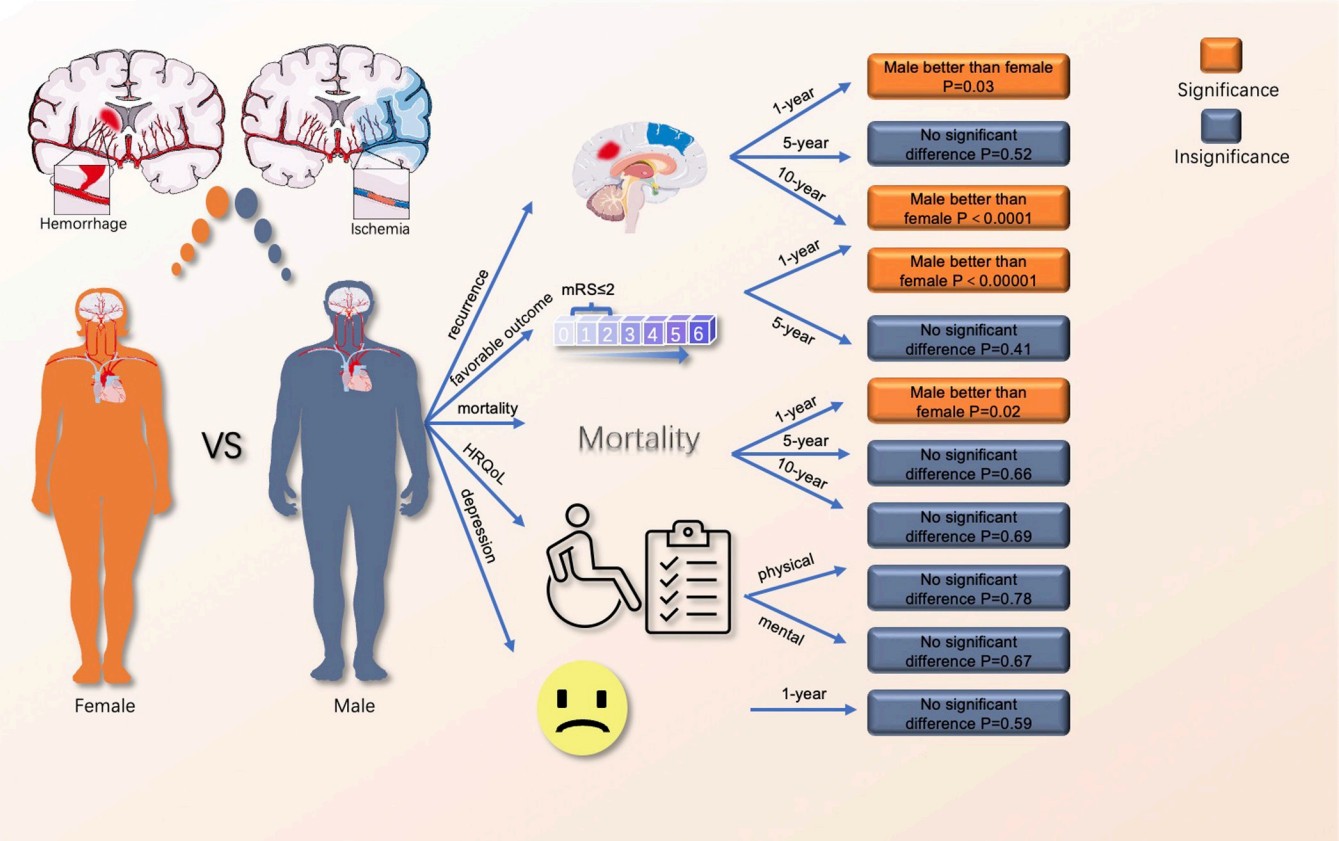

**Fig 4. The comparison of long-term outcomes between males and females.** Parts of the Fig 4 (such as human outlines and brain images) were drawn by using pictures from Servier Medical Art. Servier Medical Art by Servier is licensed under a Creative Commons Attribution 3.0 Unported License (https://creativecommons.org/licenses/by/3.0/).

In our meta-analysis, the finding that no significant difference was found in depression and health-related quality of life between male and female, should be interpreted with cautioun because of the significant heterogeneity. Furthermore, it is difficult to pool the data with consistent standards because different measurements were utilized to define the outcomes of quality of life, depression, anxiety, activities of daily living, and cognitive impairment. Therefore, future studies using a unified measurement and formula mode are warranted to further compare these outcomes between male and female patients.

Previous studies [25, 44, 45] reported inconsistent results regarding sex differences in the prognosis of patients with stroke. However, in the present meta-analysis, females tended to have poorer outcomes, higher stroke recurrence in the first year after stroke, and higher mortality in the first and tenth years. Therefore, it is necessary to focus on potential sex-based differences in managing long-term health care in patients with stroke.

This meta-analysis had some limitations. First, all studies included in the present analysis were cohort studies. Among the included studies, 15 were conducted retrospectively, and any conclusions drawn were consequently subject to the limitations of the retrospective study design, including recall and observer bias. In addition, 14 studies were hospital-based, which may have introduced selection bias. More population-based studies are warranted to further examine sex-based differences in the long-term outcomes after stroke. Furthermore, due to the lack of patient level data, further analysis including different age-groups and disease

severity could not be performed in the present meta-analysis to identify long-term outcomes between male and female. Future studies are needed to address this issue.

## 5. Conclusion

This meta-analysis found that 1- and 10-year mortality, as well as stroke recurrence rates, were higher in female patients than in male patients after stroke. In addition, females tend to experience less favorable outcomes in the first year after a stroke. Further long-term studies on sex disparities in stroke prevention, care, and management are warranted to explore the opportunities to reduce this gap.

## Supporting information

**S1 Checklist. PRISMA 2020 checklist.**
(DOCX)

**S1 File. S1 Text. Detail search query. S1 Fig. Forest plot for 1-year mortality, stroke recurrence and favorable outcome.**
(PDF)

## Author Contributions

**Conceptualization:** Xiumei Guo, Yu Xiong, Xinyue Huang, Xiaodong Kang, Chunhui Chen, Jianfeng Zhou, Hanlin Zheng, Yuping Chen, Weipeng Hu, Lingxing Wang, Feng Zheng.

**Data curation:** Xiumei Guo, Yu Xiong, Xinyue Huang, Zhigang Pan, Xiaodong Kang.

**Formal analysis:** Xiumei Guo.

**Methodology:** Xiumei Guo, Chunhui Chen.

**Software:** Xiumei Guo, Yu Xiong, Xinyue Huang, Zhigang Pan, Xiaodong Kang, Chunhui Chen, Jianfeng Zhou.

**Writing – original draft:** Xiumei Guo, Zhigang Pan, Hanlin Zheng.

**Writing – review & editing:** Xiumei Guo, Yuping Chen, Weipeng Hu, Lingxing Wang, Feng Zheng.

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
