## [Decision Letter · Decision Letter 0]

31 Jan 2023

PONE-D-23-00740Sex-based differences in long-term outcomes after stroke: a meta-analysisPLOS ONE

Dear Dr. Zheng,

Thank you for submitting your manuscript to PLOS ONE. After careful consideration, we feel that it has merit but does not fully meet PLOS ONE’s publication criteria as it currently stands. Therefore, we invite you to submit a revised version of the manuscript that addresses the points raised during the review process.

Please submit your revised manuscript by Mar 17 2023 11:59PM. If you will need more time than this to complete your revisions, please reply to this message or contact the journal office at plosone@plos.org. Please include the following items when submitting your revised manuscript:A rebuttal letter that responds to each point raised by the academic editor and reviewer(s). You should upload this letter as a separate file labeled 'Response to Reviewers'.A marked-up copy of your manuscript that highlights changes made to the original version. You should upload this as a separate file labeled 'Revised Manuscript with Track Changes'.An unmarked version of your revised paper without tracked changes. You should upload this as a separate file labeled 'Manuscript'.

We look forward to receiving your revised manuscript.

Kind regards,

Ahmed Nasreldein, MD

Academic Editor

PLOS ONE

Journal Requirements:

2. Please ensure that you include a title page within your main document. You should list all authors and all affiliations as per our author instructions and clearly indicate the corresponding author.

Reviewers' comments:

Reviewer's Responses to Questions

**Comments to the Author**

1. Is the manuscript technically sound, and do the data support the conclusions?

Reviewer #1: No

Reviewer #2: Yes

2. Has the statistical analysis been performed appropriately and rigorously? 

Reviewer #1: Yes

Reviewer #2: Yes

3. Have the authors made all data underlying the findings in their manuscript fully available?

Reviewer #1: No

Reviewer #2: Yes

4. Is the manuscript presented in an intelligible fashion and written in standard English?

Reviewer #1: Yes

Reviewer #2: Yes

5. Review Comments to the Author

Reviewer #1: The authors address the important topic of gender differences in long-term outcome after stroke. They identified twenty-two cohort studies and analysed 84538 patients, of whom with 49.8% were women.

Their results showed that women had a higher mortality after 1 and 5 years, higher rate of recurrent strokes and less favourable 1-year outcome.

The results are very important and strengthen the need for further investigations in this area of stroke medicine. The results are a basis for the development of interventional studies aiming at gender equity.

The analysis is seems well conducted however, I cannot fully review the manuscript. I was not able to read Figure 1, which shows the main results. The image quality is too poor and the figure appears too blurry. Nothing can be identified. Same for Figure 4. It was not possible to review most of the results.

I have the following comments to the introduction and methods:

Introduction:

I suggest to omit the "general" sentence that stroke is a leading cause of death and disability. This Is well known and it would be much more attracting for readers to start with the topic, e.g. start with sentence two.

The authors describe that they used the "Cochrane risk bias assessment tool was used to assess the bias in randomized controlled studies". Was this the "Cochrane risk-of-bias tool for randomized trials (RoB 2)" tool? If so, it would be easier for the reader if this was indicated in the sentence in detail.

Just out of curiosity, what was the reason to not use the ROBINS-I tool for observational studies?

From the text, I do not understand whether the sensitivity analysis for mortality was performed for 1 year and 5 years? On Page 10, lines 200-203 only one results is presented.

In the paragraph addressing the secondary outcomes depression and health related quality of live, the authors describe that there was no significant difference between men and women. This is also discussed on page 12. Considering the serious heterogeneity of the studies with I2 between 68 and 97%, the conclusion should be that there is not enough evidence to answer this question.

Is any analysis available comparing data from men and women of the same age-group and disease severity?

Reviewer #2: Dear Authors

I congratulate you on a well-written article which attempts to highlight an important yet less addressed issue in stroke. I would like to however point out few discrepancies and urge you to correct these.

1. While the results clearly state that the mortality rates were higher for women at 1 year and 10 years, repeatedly in the paper (abstract, body and conclusion) it is mentioned at 1 and 5 year significant mortality rates. I believe this is a mistake and needs to be edited.

2. Again in the results HRQoL shows no difference between males and females, however in the discussion authors have highlighted and discussed a difference in the poorer mental and physical QOL in women.

3. The consort figure also has discrepancies. The number of excluded articles with reasons do not tally together and some information is either missing or incorrect.

6. PLOS authors have the option to publish the peer review history of their article (what does this mean?). If published, this will include your full peer review and any attached files.

Reviewer #1: No

Reviewer #2: **Yes: **Ivy Anne Sebastian

---

## [Author Response · Author response to Decision Letter 0]

14 Feb 2023

February 14, 2023

Ahmed Nasreldein, MD

Academic Editor

PLOS ONE

Dear Editor Ahmed Nasreldein

Thank you so much for your e-mail dated 31 January 2022, providing yours and the Reviewers’ comments on our submission (Manuscript ID: PONE-D-23-00740) entitled “Sex-based differences in long-term outcomes after stroke: a meta-analysis”. Following your suggestions and the reviewers` comments, our manuscript has been revised completely and re-submitted, along with the point-by-point responses to the comments. 

In the re-submitted manuscript, the revised parts have been marked in yellow. We believe that the revised manuscript can meet the publication standards of your journal.

Thank you again for your help in improving our manuscript.

Yours sincerely,

Feng Zheng

Professor and Director of Department of Neurosurgery,

The Second Affiliated Hospital,

Fujian Medical University

34 Zhongshan North Road

Quanzhou 362000

Fujian Province

P.R. of China

Tel: + (86)13225985688 

Email: dr.feng.zheng@gmail.com

Reviewer 1 evaluation: The authors address the important topic of gender differences in long-term outcome after stroke. They identified twenty-two cohort studies and analysed 84538 patients, of whom with 49.8% were women. Their results showed that women had a higher mortality after 1 and 5 years, higher rate of recurrent strokes and less favourable 1-year outcome. The results are very important and strengthen the need for further investigations in this area of stroke medicine. The results are a basis for the development of interventional studies aiming at gender equity.

Q1: The analysis is seems well conducted however, I cannot fully review the manuscript. I was not able to read Figure 1, which shows the main results. The image quality is too poor and the figure appears too blurry. Nothing can be identified. Same for Figure 4. It was not possible to review most of the results.

Response to the comment: Thank you so much for your positive and constructive suggestions. Following your comments, the clear figures with better quality have been resubmitted. Please see the revised Figure 1, 2, 3, 4 below.

Figure 1 Flow chart showing the search strategy 

Figure 2 Forest plots showing primary outcomes between males and females. M-H Mantel-Haenszel statistic

Figure 3 Forest plot showing secondary outcomes between males and females. M-H: Mantel-Haenszel statistic

Figure 4 The comparison of long-term outcomes between males and females

Q2: I have the following comments to the introduction and methods:

Introduction: I suggest to omit the "general" sentence that stroke is a leading cause of death and disability. This Is well known and it would be much more attracting for readers to start with the topic, e.g. start with sentence two.

Response to the comments: Thank you so much for your suggestion. Following your idea, the "general" sentence that stroke is a leading cause of death and disability has been omitted, and the revised manuscript is started with sentence two. Please see line 67-68.

(Line 67-68, Introduction)

Stroke affects female and male patients differently, although the reasons and mechanisms of these differences are unclear (1, 2).

Q3: The authors describe that they used the "Cochrane risk bias assessment tool was used to assess the bias in randomized controlled studies". Was this the "Cochrane risk-of-bias tool for randomized trials (RoB 2)" tool? If so, it would be easier for the reader if this was indicated in the sentence in detail.

Just out of curiosity, what was the reason to not use the ROBINS-I tool for observational studies?

Response to the comments: Yes, the "Cochrane risk bias assessment tool" in our manuscript was the "Cochrane risk-of-bias tool for randomized trials (RoB 2)" you mentioned, which has been replaced accordingly. Please see line 137-138. 

Additionally, at the recommendation of Handbook for Systematic Reviews of Interventions, the Newcastle–Ottawa Scale (NOS) was adopted in this meta-analysis, which is the most popular tool for evaluating risk bias of observational cohort studies when performing meta-analysis. ROBINS-I is a new tool for evaluating risk bias in estimates of the effectiveness or safety (benefit or harm) of an intervention from studies that do not use randomized to allocate interventions. If possible, we would use this tool in our next studies.

(Line 137-138, Methods)

Cochrane risk-of-bias tool for randomized trials (RoB 2) was used to assess the bias in randomized controlled studies (17).

Q4: From the text, I do not understand whether the sensitivity analysis for mortality was performed for 1 year and 5 years? On Page 10, lines 200-203 only one results is presented.

Response to the comments: We appreciate your comment. Yes, sensitive analysis was performed for 1 year and 5 years. Due to no exact heterogeneity source detected after sensitivity analysis for the outcome of 1 year mortality, we just presented the results of sensitivity analysis for 5-year mortality. Following your idea, this issue has been corrected in the revised manuscript. Please see line 194-196 and line 254-256.

(Line 194-196, Results)

Due to the substantial heterogeneity in the assessments of above two outcomes (I2=90%, P<0.00001 and I2=92%, P<0.00001), a sensitivity analaysis was performed, without having detected the exact heterogeneity source.

(Line 254-256, Discussion)

In our meta-anlaysis, the conclusion that female patients with stroke had significantly higher 1-year mortality rates should be interpreted with caution because of the significant heterogeneity.

Q5: In the paragraph addressing the secondary outcomes depression and health related quality of live, the authors describe that there was no significant difference between men and women. This is also discussed on page 12. Considering the serious heterogeneity of the studies with I2 between 68 and 97%, the conclusion should be that there is not enough evidence to answer this question.

Response to the comments: Thank you for your constructive suggestion. There was no significant difference between men and women, and we have revised this conclusion accordingly. Please see line 277-279.

(Line 277-279, Discussion)

In our meta-analysis, the finding that no significant difference was found in depression and health-related quality of life between male and female, should be interpreted with cautioun because of the significant heterogeneity.

Q6: Is any analysis available comparing data from men and women of the same age-group and disease severity?

Response to the comments: Thank you so much for your constructive comments. Due to the lack of patient level data comparing long-term stroke outcomes between male and female, further analysis cannot be achieved in the present analysis, which has been addressed in the limitation of the revised manuscript. Please see line 294-296 below.

(Line 294-296, Discussion)

Furthermore, due to the lack of patient level data, further analysis including different age-groups and disease severity could not be performed in the present meta-analysis to identify long-term outcomes between male and female. Future studies are needed to address this issue.

Reviewer 2 evaluation: I congratulate you on a well-written article which attempts to highlight an important yet less addressed issue in stroke. I would like to however point out few discrepancies and urge you to correct these.

Q1: While the results clearly state that the mortality rates were higher for women at 1 year and 10 years, repeatedly in the paper (abstract, body and conclusion) it is mentioned at 1 and 5 year significant mortality rates. I believe this is a mistake and needs to be edited.

Response to the comments: Thank you for your constructive and positive comments. Yes, this is a mistake, which has been corrected accordingly. Please see line 58, line 301 below.

(Line 58, Abstract)

In this meta-analysis, the 1- and 10-year mortality and stroke recurrence rates were higher in female patients than in male patients after stroke. 

(Line 301, Conclusion)

This meta-analysis found that 1- and 10-year mortality, as well as stroke recurrence rates, were higher in female patients than in male patients after stroke.

Q2: Again in the results HRQoL shows no difference between males and females, however in the discussion authors have highlighted and discussed a difference in the poorer mental and physical QOL in women.

Response to the comments: Thank you for your comments. Yes, there is no significant difference in the outcomes of mental and physical QOL between male and female. The issue mentioned has been revised accordingly. Please see line 277-279 below.

(Line 277-279, Discussion)

In our meta-analysis, the finding that no significant difference was found in depression and health-related quality of life between male and female, should be interpreted with cautioun because of the significant heterogeneity.

Q3: The consort figure also has discrepancies. The number of excluded articles with reasons do not tally together and some information is either missing or incorrect.

Response to the comments: Thank you for your constructive comments. Following your suggestions, the number of excluded articles with reasons, along with the missing or incorrect information, has been revised. Please see line 174-178, Figure 1 below.

(Line 174-178, Results)

The full text of the remaining 227 articles was assessed for eligibility, resulting in the exclusion of 150 studies with outcomes not stratified by sex and 37 conference abstracts without a complete text. Three studies were excluded because their data were not extractable. Fifteen of the remaining articles were excluded due to a lack of long-term outcome data based on sex-based differences.

(Figure 1 Flow chart showing the search strategy)

---

## [Decision Letter · Decision Letter 1]

6 Mar 2023

Sex-based differences in long-term outcomes after stroke: a meta-analysis

PONE-D-23-00740R1

Dear Dr. Zheng,

We’re pleased to inform you that your manuscript has been judged scientifically suitable for publication and will be formally accepted for publication once it meets all outstanding technical requirements.

Kind regards,

Ahmed Nasreldein, MD

Academic Editor

PLOS ONE

Additional Editor Comments (optional):

Reviewers' comments:

Reviewer's Responses to Questions

**Comments to the Author**

1. If the authors have adequately addressed your comments raised in a previous round of review and you feel that this manuscript is now acceptable for publication, you may indicate that here to bypass the “Comments to the Author” section, enter your conflict of interest statement in the “Confidential to Editor” section, and submit your "Accept" recommendation.

Reviewer #1: All comments have been addressed

Reviewer #2: All comments have been addressed

2. Is the manuscript technically sound, and do the data support the conclusions?

Reviewer #1: Yes

Reviewer #2: Yes

3. Has the statistical analysis been performed appropriately and rigorously? 

Reviewer #1: Yes

Reviewer #2: Yes

4. Have the authors made all data underlying the findings in their manuscript fully available?

Reviewer #1: Yes

Reviewer #2: Yes

5. Is the manuscript presented in an intelligible fashion and written in standard English?

Reviewer #1: Yes

Reviewer #2: Yes

6. Review Comments to the Author

Reviewer #1: (No Response)

Reviewer #2: The authors have addressed all the suggested comments. Congratulations on a Well written article on an important topic.

7. PLOS authors have the option to publish the peer review history of their article (what does this mean?). If published, this will include your full peer review and any attached files.

Reviewer #1: No

Reviewer #2: No

---

## [Editor Report · Acceptance letter]

14 Apr 2023

PONE-D-23-00740R1 

Sex-based differences in long-term outcomes after stroke: a meta-analysis 

Dear Dr. Zheng:

I'm pleased to inform you that your manuscript has been deemed suitable for publication in PLOS ONE. Congratulations! Your manuscript is now with our production department. 

Kind regards, 

on behalf of

Dr. Ahmed Nasreldein 

Academic Editor

PLOS ONE